# Ionic Mass Transfer at Point Electrodes Located at Cathode Support Plate in an Electrorefining Cell in Presence of Rectangular Turbulent Promoters

**Tondepu Subbaiah [1,*], Ponnam Vijetha [1], Barsha Marandi [2], Kali Sanjay [2] and Manickam Minakshi [3,*]**

1   Department of Chemical Engineering, VFSTR University, Vignan's Foundation for Science, Technology and Research, Guntur 522213, India; pv_chem@vignan.ac.in
2   Hydro & Electro Metallurgy Division, Institute of Minerals and Materials Technology (Council of Scientific and Industrial Research), Bhubaneswar 751013, India; barsham@immt.res.in (B.M.); kalisanjay@gmail.com (K.S.)
3   College of Science, Health, Engineering and Education, Murdoch University, Perth, WA 6150, Australia
*   Correspondence: tsubbaiah@yahoo.com (T.S.); minakshi@murdoch.edu.au (M.M.)

**Abstract:** Current density plays a major role in deciding the plant size, current efficiency, and energy consumption in electrorefining cells. In general, operating current density will be 40% of the limiting current density. Forced circulation of the electrolyte in the presence of promoters improves the mass transfer coefficient. In the present study, rectangular turbulence promoters are fitted at the bottom side of the cell to improve the mass transfer coefficient at the cathode support plate. The limiting current density technique is used to measure the mass transfer coefficient. The variables covered in the present study are the effects of flow rate, promoter height, and spacing among the promoters. The electrolyte consists of copper sulfate and sulphuric acid. At a regulated flow rate, the electrolyte is pumped from the recirculation tank to the cell through an intermediate overhead tank. The limiting current density increased with an increasing flow rate in the presence of promoters, and thus the overall mass transfer coefficient on the cathode support plate also improved. With an increase in the flow rate of the electrolyte from $6.67 \times 10^{-6}$ to $153.33$ m$^3$/s, limiting current density increased from $356.8$ to $488.8$ A/m$^2$ for spacing of $0.30$ m, with a promoter height of $0.01$ m. However, it is noteworthy that when the promoter height is increased from $0.01$ to $0.07$ m, the overall mass transfer coefficient is found to increase up to 60%, but with the further increase in the promoter height to $0.30$ m the mass transfer coefficient starts to decrease. Therefore, the optimized cell parameters are established in this work. The current sustainable concept of employing rectangular turbulence promoters will bring benefits to any precious metal refining or electrowinning tank house electrolytes.

**Keywords:** limiting current density; mass transfer; electro-refining; flow rate; velocity; electrowinning; correlations

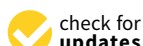



## 1. Introduction

Electrochemical processes include electrodeposition/electrowinning, and electrorefining techniques, in which precious metals are either recovered or purified from the solutions containing cations of the metal in an electrolytic cell [1,2]. Typically, copper extracted from ore receives an electrolytic treatment either by electrorefining or electrowinning processes. Electrorefining is much more commonly practiced than electrowinning, and such plants are built throughout the world. Electrorefining produces blister copper (impure copper), which must be refined before use. Electrorefining consists of electrochemically dissolving copper from impure anodes and selectively depositing the copper in pure form onto copper cathodes through electrolysis. The anodes and cathodes are suitably designed to produce high-quality copper at relatively higher current densities in copper refining and electrowinning units. Since electrochemical reactions occur only at electrode interfaces,

the current is directly proportional to the area of the interfaces and therefore the current is usually normalized to a unit area and expressed in current density [1–4]. As such, in typical electrorefining plants, current density performs an important role in determining initial investment, floor area, current efficiency, and energy consumption. Nevertheless, increasing the current density outside its threshold termed "limiting current density" produces undesirable rough and unclean deposits. Consequently, the appropriate selection of current density plays a significant role in deciding the operating conditions of the electrolytic cell and in the output of the metal deposits. Higher current density can be obtained by applying forced convection. However, similar advantages can be obtained by cell operation at comparatively lower flow rates by applying the turbulence promoters.

The maximum output with reduced cost can be achieved by increasing the ionic mass transfer rates by using various types of augmentation methods such as forced convection of the electrolyte, use of insert promoters, crossflow elements in the flow streams, solids presence in a packed or fluidized state, vibration or rotation of the surface in stationary or flowing electrolyte, pulsation of the electrolyte, and agitation induced by the stirring of the electrolyte. Among the various augmentation methods specified, the use of turbulence promoters is widespread; the energy losses associated with it are of a high order but yield higher mass transfer coefficients by several folds compared to that of smooth flow. The use of promoters is not only connected to the copper refinery system but is also commonly deployed in other electrorefining plants worldwide. Therefore, the use of turbulence promoters is suitable for any refining or electrowinning plant system.

Several authors [5–18] have studied the augmentation in ionic mass transfer rates by introducing turbulence promoters in the flow path of the electrolyte and reported improving the performance of the system. A very brief literature review reported in the past and recently has been outlined here.

According to comprehensive research performed in the past for turbulence promoters, initially Leitz and Marincic [8] employed rectangular, triangular, and circular promoters adjacent to the wall of a parallel piped electrochemical cell and identified the best geometric type of promoters based on the obtained overall mass transfer data. Blum and Oliver [9], and Gambill and Greene [10], reported that using an inlet vertex flow generator improved the mass transfer, whereas Nieva and Bohm [11] employed staggered tube banks of equilateral triangle arrangements for local mass transfer studies. Later, Krishna et al. [12] studied ionic mass transfer in the presence of turbulence promoters in a circular conduit and demonstrated its improvement in mass transfer coefficients up to 6.5. Subsequently, Venkateswarlu [13] conducted a study on coaxially positioned disks on a rod as turbulence promoters for a similar ionic mass transfer study. Kang and Chang [14] studied promoters of different geometric shapes of zigzag and cavity types that were fixed at lower or upper walls. Sedahmed and Shemilt [15] attempted to augment mass transfer coefficients between an electrolyte and horizontal, vertical, and inclined cylinders for the deposition of copper. Likewise, Stork and Hutin [16] used turbulence promoters for the recovery of copper from dilute solutions. To this end, ionic mass transfer studies in presence of turbulence promoters using the electrorefining technique are widely reported [16–18] for enhanced performance. All these previous studies inspired Subbaiah et al. [17] to study mass transfer conditions at centrally located electrodes at a cathode support plate in copper refinery cells using plate-type turbulence promoters. Further to this, Subbaiah et al. [18] made a further modification to the earlier version and reported an enhanced mass transfer using vertical plate turbulence promoters.

Recently, semi-circular teethed promoters were reported by Sameera et al. [19] to be employed at the bottom of the cell, and the correlation between mass transfer coefficient and spacing between the promoters has been studied. Kharicha et al. [20] reported that the variation in the intensity of the flow of electrolytes impacts the concentration field that determines the mass transfer at interfaces. Then, to enhance the mass transfer and the overall performance of the reaction kinetics, Gallent et al. [21] introduced a pulsed electrolyte flow in the electrolytic cell, which resulted in an improved mass transfer to some

extent. Kubanek and Krewer [22] studied the interaction of electrochemical reactions with transport phenomena and double-layer charging by species frequency response analyses. The effect of air pressure and mesh promoters in the upgraded electrochemical cell design has been found [23] to provide mass transfer benefits from perpendicular gas flow incoming in the cell. Another study by Chandralekha et al. [24] showed that mass transfer coefficients decrease as the height increases and depend on the characteristic length of the promoter, typically a single pentagonal plate assembly. All these previous studies in the past and in present invariably suggest that electrochemical reaction kinetics are often influenced by electrolytic cell design for industrial production.

In the present study, rectangular promoters were employed, and their sustainable concepts, which could be beneficial for electrorefining tank house electrolytes in terms of operation feasibility, and cost-effectiveness, are reported. In general, both the rectangular and cylindrical promoters will enhance the limiting current density while improving the mass transfer coefficients. However, we found that the extent of variation is found to be maximum for the current rectangular set-up due to the change of flow pattern. It is worth noting that current density in the electrolytic cells can be increased in the presence of rationally designed promoters without compromising the quality of the deposits. With the increase of mass transfer coefficient, we can increase the operating current density of the plant, which is, in general, 40% of the limiting current density.

The overall aim of the current work is the improvement of the ionic mass transfer coefficient of point electrodes located at the cathode support plate at specified locations due to the presence of rectangular promoters, which has been successfully demonstrated to lead to more sustainable and cleaner production of precious metals such as copper.

## 2. Experimental

The experimental setup (Figure 1) consists of a recirculation tank (1), Watson and Marlow metering pump (2), overhead storage tank 50 L capacity (3), rotameter (4), electrolytic cell (5), current measuring equipment such as EG&G Princeton potentiostat Model 173 (6), scan generator (7), Hewlett Packard XY-recorder model 7046 A (8), selector switch (9), working electrode (10), counter electrode (11), and reference electrode (12), and control valves $V_1$ and $V_2$. The electrolytic cell is made up of fiberglass-reinforced plastic, with a length of 0.594 m, a width of 0.19 m, and a height of 0.23 m, and it is a prototype of an industrial-scale copper electrorefining cell. The dimensions of the cell are 1/6th of the industrial copper refining cell. The electrolytic cell with the location of promoters is shown in Figure 2. The anode is a 0.002 m-thick copper sheet with a length 0.18 m and a width 0.15 m, and it was rigidly suspended vertically in the cell at 0.05 m away from the exit end. As a reference electrode, a copper rod with a diameter of 0.003 m and a length of 0.05 m was immersed in the copper sulfate solution same as the electrolyte. The working electrode is a perspex plate of 0.5 cm thickness, 0. 18 m length, and 0.15 m width, on which point electrodes (0.005 m) are flush-fitted with its surface on the cathode support plate (Figure 3). The space between the point electrodes is 0.025 m vertically, and there is 0.06 m between the two columns of point electrodes. The electrolyte consists of copper sulfate (0.1 Kg·moles·m$^{-3}$) and sulfuric acid (1.0 Kg·moles·m$^{-3}$), of analytical reagent grade from E. Merck. At a regulated flow rate, the electrolyte is pumped from the recirculation tank to the cell through an intermediate overhead tank using a rotameter. The limiting current measurements were made similar to those explained in one of our previous reports [18]. After the flow rate and temperature were stabilized, the potential was applied from the potentiostat at a constant scan rate of 5 mV·s$^{-1}$ until the limiting current was approached. The current and potential were recorded with the help of an X-Y recorder. The current–potential curve shows the plateau where the current was found to be independent of the potential. This value of the current corresponds to the limiting current. The rectangular promoters (with dimensions H: 0.01 m, 0.03 m, 0.05 m, and 0.07 m) were made from 0.003 mm PVC sheets rigidly fixed at the bottom of the cell at desired spacing (S: 0.075 m, 0.15 m, and 0.30 m).

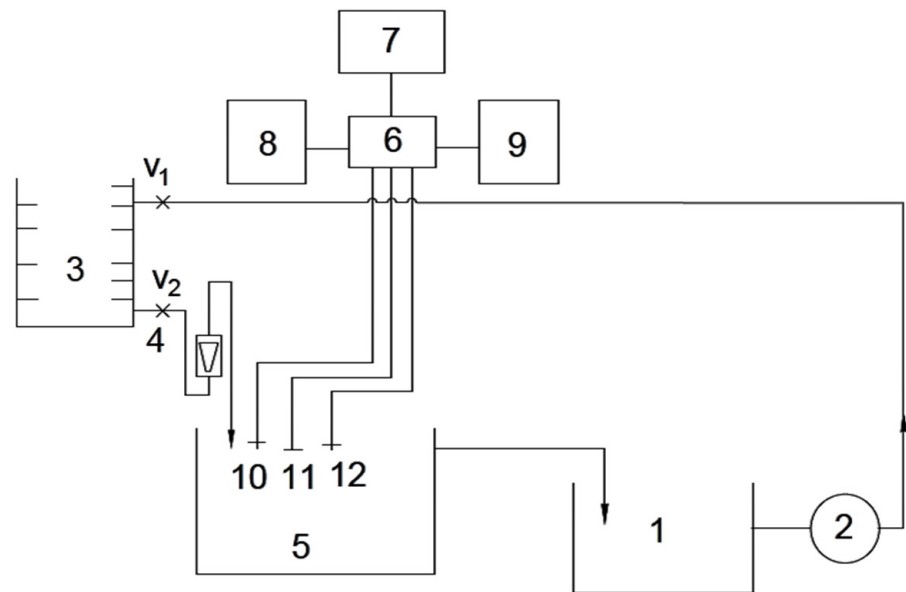

**Figure 1.** Schematic diagram of the experimental set-up.

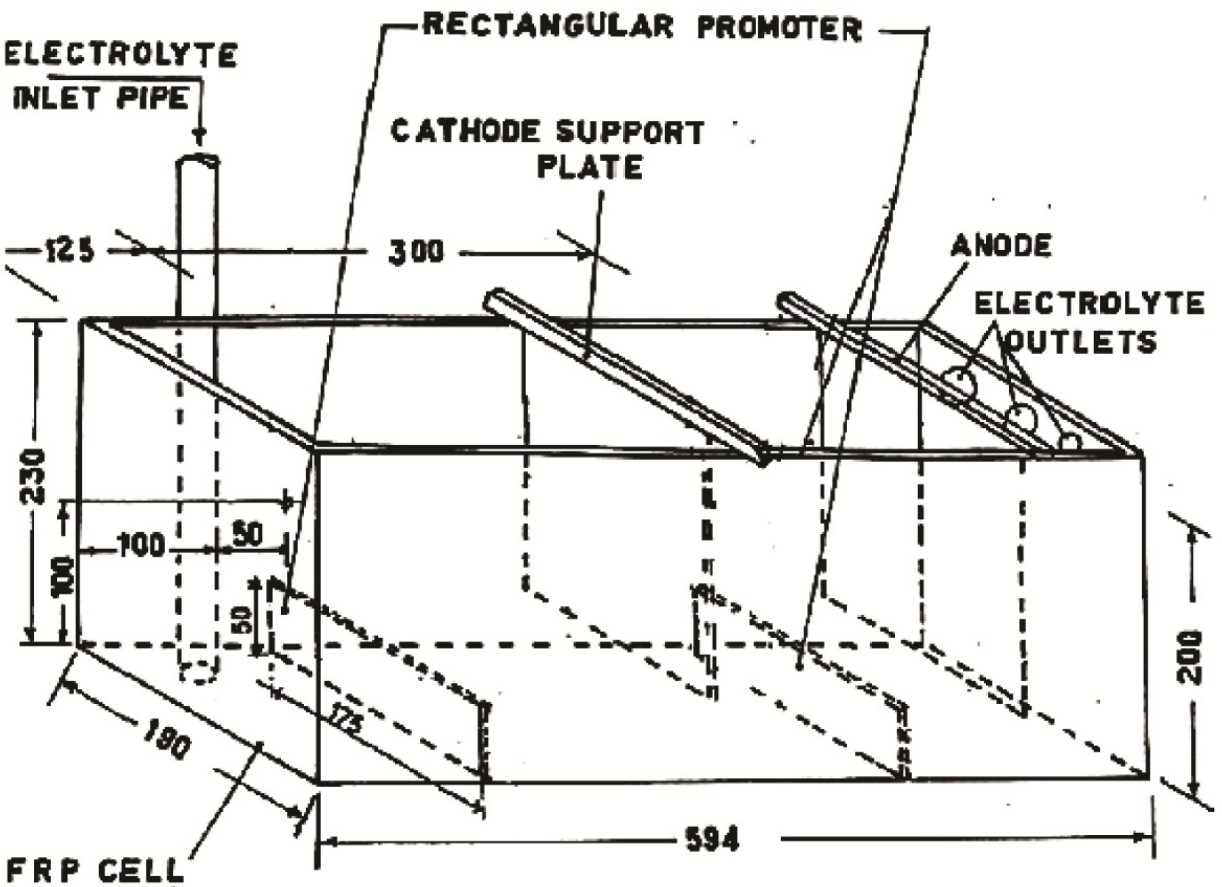

**Figure 2.** Electrolytic cell with location of promoters.

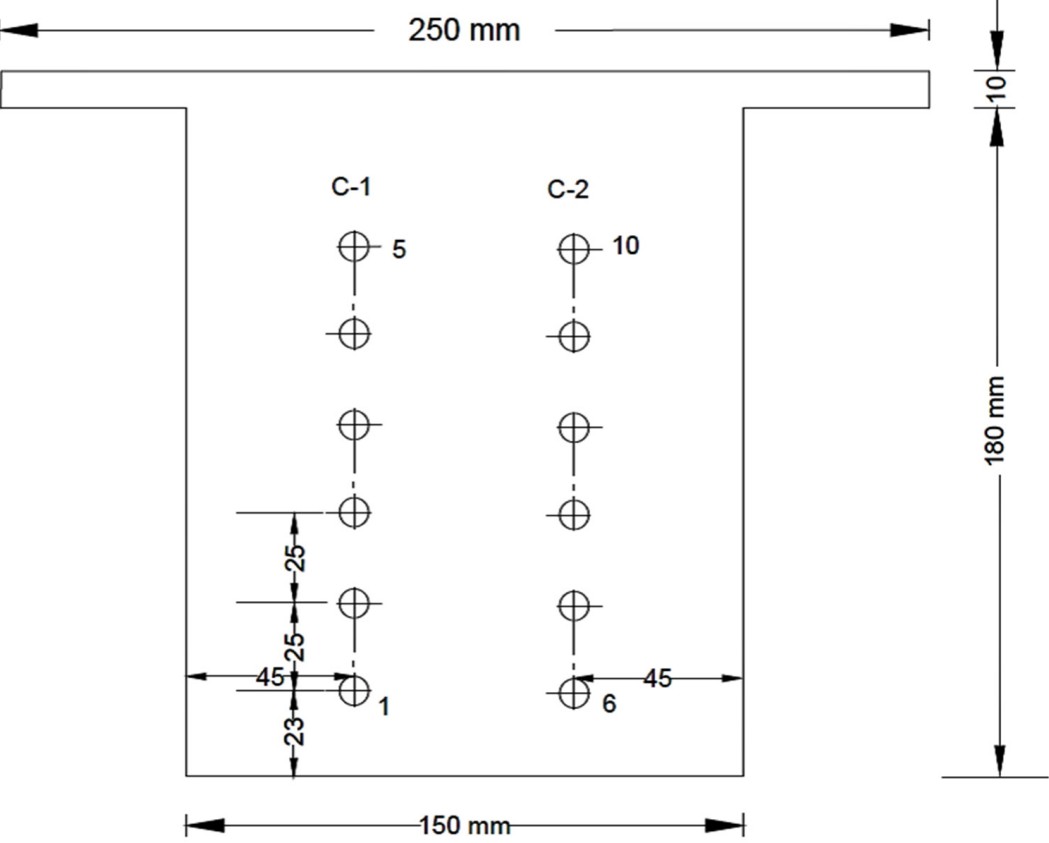

**Figure 3.** Cathode support plate with micro-electrodes.

## 3. Results and Discussion

A critical review of the literature revealed that by the use of inserts with different dimensions, such as cylinders [25], cones [26], discs, and spheres [27], the mass transfer coefficient increases several-fold depending upon the operating variables covered. In our current study, the rectangular promoter has been employed. The experimental data collected in the current set-up is based on 2600 limiting current measurements covering the S/H (space/height of the prompter) values 1 to 30 with the flow rate $21 \times 10^{-6}$ to $171 \times 10^{-6}$ m$^3$/s. Therefore, the limiting current data have been collected at different flow rates, different promoter heights, and different promoter spacings. Each limiting current density is an average of 10 limiting current densities of point electrodes. However, the primary objective of the current work is given to improve the limiting current density by using turbulence promoters. Overall, the effect of flow rate, promoter sizes, and promoter spacing has been studied.

The flow of the electrolyte through the cell in the present system can be assumed to be an open channel cell. The turbulence promoters placed at the bottom of the cell cause secondary flow locally and upwardly through the electrolyte. The cathode support plate and anode support plate create crossflow. Under these conditions, the mixing of the electrolyte can be viewed as follows:

(i)    Ripple flow-induced due to promoters;
(ii)   Crossflow due to working/counter electrode plate; and
(iii)  Slit flow caused due to narrow gap between the cathode support plate and cell wall.

The flow patterns are the possible cause for intense mixing in the cell, and mass transfer coefficients are extremely complex and not easily amenable to mathematical modeling.

The following variables were considered for measurements:

(a)    Electrolyte flow rate (Q);

(b)　Promoter height (H); and

(c)　Distance between the promoters (S).

The point electrodes were placed column-wise in five rows and numbered 1 to 10 of the cathode supporting plate. Each column contains 5 electrodes at columns 1 and 2. It is found that limiting current densities at each electrode causes fluctuation due to flow interactions and narrow variations in their values. Therefore, the arithmetic average of the limiting current densities of 10 local values obtained at individual electrodes of columns on the cathode support plate appears to be suited to further analysis. The typical average mass transfer coefficient is $2.1011 \times 10^{-5}$ m/s.

The involvement of the surplus indifferent electrolyte reduces the responding species effect on ionic mobility. Hence, the reaction to the electrode becomes regulated by diffusion. More details on the estimates made on the mass-transfer coefficient can be obtained from one of our previous reports [18]. Limiting current data have been reported at the cathode for the cathodic reduction of the cupric ion over diffusion-controlled constraints, and electrochemical reactions at the cathode (Equation (1)) and gaseous oxygen at the anode (Equation (2)) are as follows:

$$Cu^{2+} + 2e^- \rightarrow Cu^0 \tag{1}$$

$$H_2O \rightarrow \frac{1}{2} O_2 + 2H^+ + 2e^- \tag{2}$$

Data for limiting current density from reported limiting currents were estimated, and Equation (3) was used for determining the mass transfer coefficients.

$$K = I_L / n \, A \, FC \tag{3}$$

where

K = mass transfer coefficient m/s;

$I_L$ = limiting current density;

C = concentration of the reacting ion, k·mol/m$^3$;

A = area of the electrode, m$^2$;

F = Faraday's constant, 96,500 coulombs/mol.; and

n = number of electrons taking place in the reaction

### 3.1. Effect of Flow Rate

The overall limiting current density of the point electrodes located at the cathode support plate data is plotted with respect to flow rate and given in Figure 4. The data have been provided for three sizes of promoters (S = 0.30 m, H = 0.01 m; S = 0.15 m, H = 0.03 m; and S = 0.075 m, H = 0.075 m). Interestingly, as observed from the plot in Figure 4, the flow rate of the electrolyte is found to be increased from $26.67 \times 10^{-6}$ to $153.33 \times 10^{-6}$ m$^3$/s. The increase of flow rate increases the limiting current density from 356.4 to 488.8 A/m$^2$ for S = 0.30 m and H = 0.01 m. The same trend is also observed in other spacing and promoter heights, and the results are tabulated in Table 1. Table 1 shows the overall coefficient of mass transfer data against velocity for three promoters (S = 0.30 m, H = 0.01 m; S = 0.15 m, H = 0.03 m; and S = 0.075 m, H = 0.07 m). The overall mass transfer coefficient ($K_L$) is found to increase with the increase of velocity, and its variation is with a slope of $0.46 \times 10^{-2}$. The relationship is given in Equation (4).

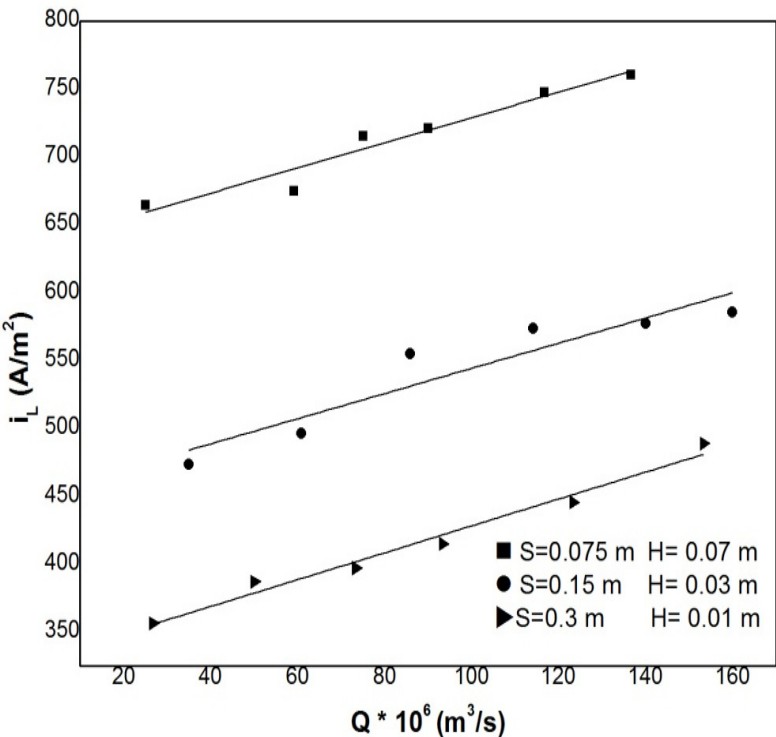

**Figure 4.** Augmentation of limiting current density at cathode support plate with increase of flow rate.

**Table 1.** (**a**) Effect of velocity on mass transfer coefficient at S = 0.075 m and H = 0.07 m. (**b**) Effect of velocity on mass transfer coefficient at S = 0.30 m and H = 0.01 m. (**c**) Effect of velocity on mass transfer coefficient at S = 0.075 m and H = 0.07 m.

| | (a) | |
|---|---|---|
| | $V \times 10^2$ m/s | $K_L \times 10^5$ m/s |
| 1 | 0.0658 | 3.1338 |
| 2 | 0.1559 | 3.1819 |
| 3 | 0.1973 | 3.3733 |
| 4 | 0.2368 | 3.3993 |
| 5 | 0.3070 | 3.5253 |
| 6 | 0.3596 | 3.5870 |
| | (b) | |
| | $V \times 10^2$ m/s | $K_L \times 10^5$ m/s |
| 1 | 0.0702 | 1.7357 |
| 2 | 0.1316 | 1.8848 |
| 3 | 0.1930 | 1.9335 |
| 4 | 0.2450 | 2.0206 |
| 5 | 0.3245 | 2.1697 |
| 6 | 0.4034 | 2.3806 |

**Table 1.** *Cont.*

| | (c) | |
|---|---|---|
| | $V \times 10^2$ m/s | $K_L \times 10^5$ m/s |
| 1 | 0.0921 | 2.3604 |
| 2 | 0.1600 | 2.4762 |
| 3 | 0.2256 | 2.7680 |
| 4 | 0.3000 | 2.8607 |
| 5 | 0.3678 | 2.8782 |
| 6 | 0.4203 | 2.9216 |

$$K_L = 0.46 \times 10^{-2} \, V \qquad (4)$$

where

$K_L$ = overall mass transfer coefficient, m/s;
V = velocity of the electrolyte based on the equivalent diameter of the cell, m/s.

### 3.2. Effect of Promoter Height

The dependence of promoter height on the overall coefficients is shown in Figures 5 and 6. The respective plots illustrate the data with promoters of different heights (H = 0.01 m, 0.03 m, and 0.07 m) at two spacings: S = 0.30 m and S = 0.075 m. The results indicate that with an improvement in the promoter's height, the overall coefficient improved, with an improvement of up to 60% across the height variability. The observed trend is similar to that reported by Sameera et al. [19]. The improvement in promoter height triggered the disruption to the trends of flow, which continued to spread upward. This caused the local velocities to cross the support plate for the cathode and contributed to a rise in the coefficients of overall mass transfer.

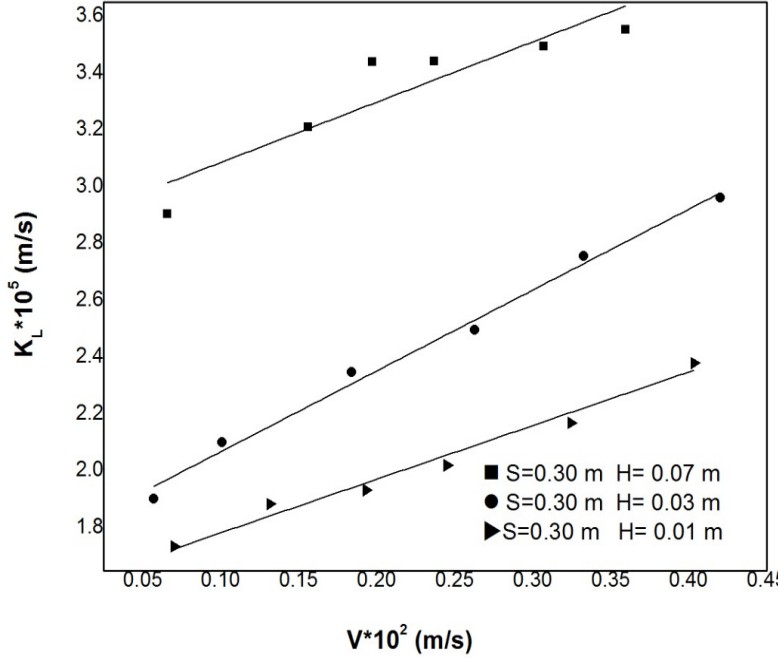

**Figure 5.** Effect of promoter height on overall mass transfer coefficients at S = 0.30 m.

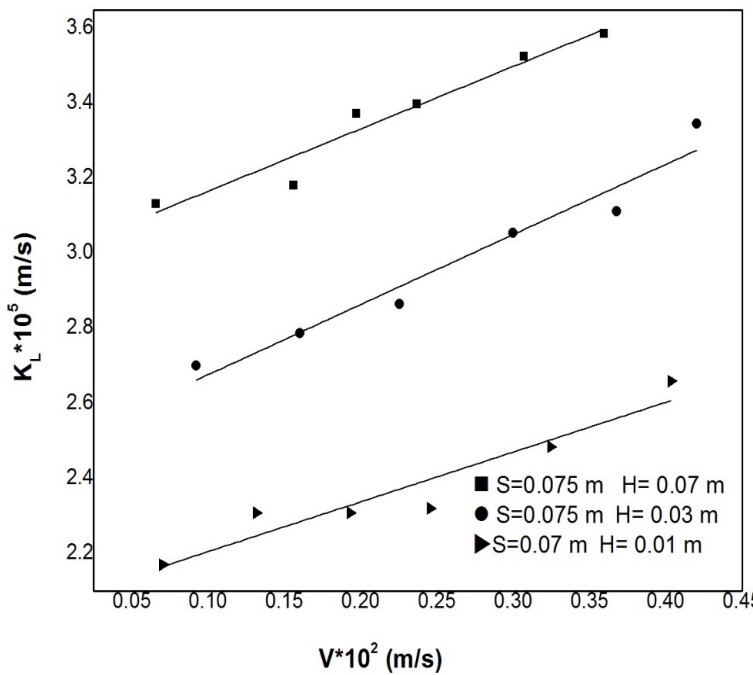

**Figure 6.** Effect of promoter height on overall mass transfer coefficients at S = 0.075 m.

### 3.3. Effect of Promoter Spacing

In Figure 7, for three promoter spacings at different promoter heights (H = 0.01 m, H = 0.03 m, and H = 0.07 m), the overall mass-transfer coefficients can be seen. It is observed in Figure 7 that with the variation in promoter spacing, the overall coefficients were found to differ inversely. The ripple flow conversed with the axial flow, which passed through the cathode support plate (CSP) at lower spacing. Additionally, the CSP served as an obstacle to axial flow. The cumulative outcome of these two results culminated in stronger instability, leading to higher coefficients. Nevertheless, a slow decay in the ripple flow could have appeared at higher spacings as it traveled across the other promoter component.

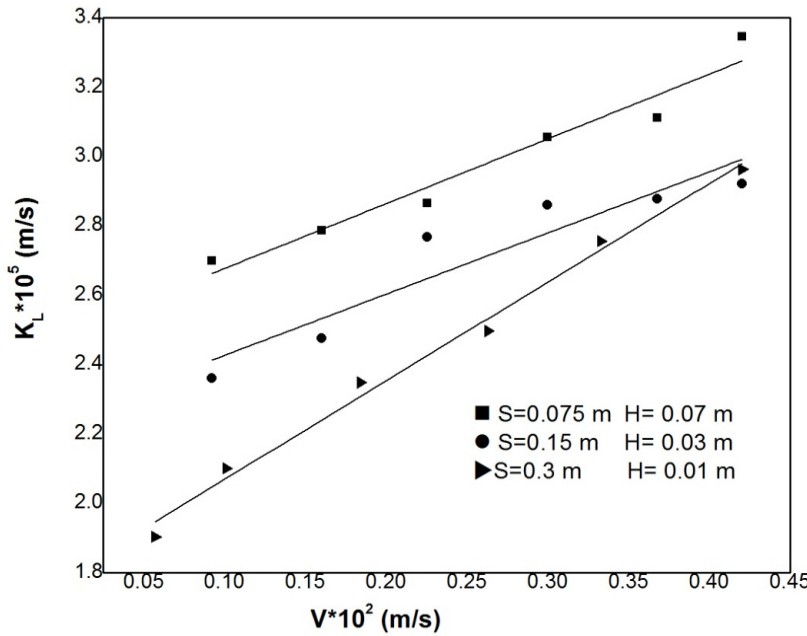

**Figure 7.** Effect of promoter spacing on overall mass transfer coefficient.

## 4. Conclusions

1. With the increase of flow rate of the electrolyte from $26.67 \times 10^{-6}$ to $153.33 \times 10^{-6}$ m$^3$/s, limiting current density increased from 356.8 to 488.8 A/m$^2$ for spacing of 0.30 m and promoter height of 0.01 m.

2. With the increase of velocity from $0.0658 \times 10^{-2}$ m/s to $0.3596 \times 10^{-2}$ m/s, mass transfer coefficient increased from $3.1338 \times 10^{-5}$ m/s to $3.5870 \times 10^{-5}$ m/s for spacing of S = 0.075 m and promoter height 0.07 m. A similar trend was also observed in all other cases of promoter spacing and promoter height. The overall mass transfer coefficient was found to increase with the increase of velocity, and its variation was with a slope of $0.46 \times 10^{-2}$.

3. With the increase of promoter height from 0.01 m to 0.07 m, the improvement of the overall mass transfer coefficient was as high as 60%.

4. With a reduction in the promoter spacing, the overall mass transfer coefficient increased. The cumulative effect of radial flow and axial flow resulted in higher mass transfer coefficients.

**Author Contributions:** Conceptualization, T.S. Methodology, investigation, T.S. and K.S. Writing—original draft preparation, P.V. and B.M. Writing—review and editing, and formal analysis, T.S. and M.M. Project administration, P.V. and B.M. All authors have read and agreed to the published version of the manuscript.

**Funding:** This research received no external funding.

**Institutional Review Board Statement:** Not applicable.

**Informed Consent Statement:** Not applicable.

**Data Availability Statement:** Not applicable.

**Acknowledgments:** This work was supported by the Vignan's Foundation for Science, Technology, and Research, India.

**Conflicts of Interest:** The authors declare that they have no known competing financial interests or personal relationships that could have influenced the work reported in this paper.

## Nomenclature

| | |
|---|---|
| A | area of the electrode, m$^2$ |
| CO | concentration of the reacting ion, kmol/m$^3$ |
| D$_e$ | equivalent diameter of the electrolytic cell, 4 Wh/(W + 2 h), m |
| D$_L$ | diffusivity, m/s |
| F | Faraday, 96,500 Coulombs/equivalent |
| h | height of the electrolyte in the cell, m |
| H | height of the promoter, m |
| I | limiting current, A |
| I$_L$ | limiting current density, A/m$^2$ |
| K$_L$ | overall mass transfer coefficient, m/s |
| Q | flow rate, m$^3$/s |
| S | spacing of the promoter, m |
| V | velocity of the electrolyte based on the equivalent diameter of the cell. |
| W | width of the electrolytic cell, m |
| p | density of the electrolyte, Kg/m$^3$ |
| n | number of electrons taking place in the reaction |

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
