# Peer review of "Ionic Mass Transfer at Point Electrodes Located at Cathode Support Plate in an Electrorefining Cell in Presence of Rectangular Turbulent Promoters"

_sustainability, doi:10.3390/su14020880_

Round 1

Reviewer 1 Report

Authors have taken into account my comments. I agree for acceptance of this article. 

Author Response

Dear Reviewer,

Many thanks for taking the time to consider our manuscript and providing the outcome. Much appreciated.

Manickam

Reviewer 2 Report

The paper is fairly modified based on my earlier comments and I recommend it for the acceptance in the current form.

Author Response

Dear Reviewer #2,

Many thanks for taking the time to consider our manuscript and providing the outcome. Much appreciated.

Manickam

Reviewer 3 Report

The present Project Report entitled “Ionic Mass Transfer at Point Electrodes Located at Cathode Support Plate in an Electrorefining Cell in Presence of Rectangular Turbulent Promoters” Tondepu Subbaiah et al., describe the rectangular turbulence promoters are fitted at the bottom side of the cell to improve the mass transfer coefficient at the cathode support plate. The limiting current density technique is used to measure the mass transfer coefficient. Furthermore, the electrolyte consists of copper sulfate and sulphuric acid. At a regulated flow rate, the electrolyte is pumped from the recirculation tank to the cell through an intermediate overhead tank and also the limiting current density increased with an increasing flow rate in the presence of promoters, and thus the overall mass transfer coefficient on the cathode support plate also improved. The authors report an interesting approach and also, I observed that they answered all the queries raised by the previous reviewers. However, certain Minor issues are detailed below which need to be addressed before its final acceptance in Sustainability.

I advise the authors to take the following points into account while revising their manuscript.

Comment 1:  There are so many typographical errors in the manuscript text, so authors need to correct them in the revised manuscript.

Comment 2: The abstract is poorly written, should be edited. It must summarize well the obtained results.

Comment 3: In the introduction section, the authors need to add some recent literature to strengthen the Introduction section.

Comment 4: All Figure's quality is very bad, so please enhance all figures or provide high-resolution figures.

Comment 5: In the whole manuscript the authors must be taken care of the superscripts and subscripts and abbreviations and also the equations are not arranged properly and they are overlapping one on another. So, correct it maintain the consistency

Comment 6: Conclusions section is written poorly, should be enhanced properly. 

Comment 7: The references style is not maintained uniformly, so please refer to the MDPI standard reference style and correct all the references accordingly.

Author Response

Dear Reviewer #3,

Many thanks for taking the time to consider our manuscript and making possible these very constructive comments.

Please find below a point-by-point response to your comments. The changes are incorporated in the revised version.

Thank you.

Manickam
